# Cellular and Molecular Mechanisms of Pathogenic and Protective Immune Responses to SARS-CoV-2 and Implications of COVID-19 Vaccines

**DOI:** 10.3390/vaccines11030615

**Published:** 2023-03-08

**Authors:** Sheikh Mohammad Fazle Akbar, Mamun Al Mahtab, Sakirul Khan

**Affiliations:** 1Department of Gastroenterology and Metabology, Ehime University Graduate School of Medicine, Toon 791-0295, Ehime, Japan; 2Interventional Hepatology Division, Department of Hepatology, Bangabandhu Sheikh Mujib Medical University, BSMMU, Dhaka 1000, Bangladesh; 3Department of Microbiology, Faculty of Medicine, Oita University, Yufu 879-5593, Oita, Japan

**Keywords:** SARS-CoV-2, COVID-19, immunity, pathogenic, protective

## Abstract

Severe acute respiratory syndrome coronavirus 2 (SARS-CoV-2) infection has devastated the world with coronavirus disease 2019 (COVID-19), which has imparted a toll of at least 631 million reported cases with 6.57 million reported deaths. In order to handle this pandemic, vaccines against SARS-CoV-2 have been developed and billions of doses of various vaccines have been administered. In the meantime, several antiviral drugs and other treatment modalities have been developed to treat COVID-19 patients. At the end of the day, it seems that anti-SARS-CoV-2 vaccines and newly developed antiviral drugs may be improved based on various new developments. COVID-19 represents a virus-induced, immune-mediated pathological process. The severity of the disease is related to the nature and properties of the host immune responses. In addition, host immunity plays a dominant role in regulating the extent of COVID-19. The present reality regarding the role of anti-SARS-CoV-2 vaccines, persistence of SARS-CoV-2 infection even three years after the initiation of the pandemic, and divergent faces of COVID-19 have initiated several queries among huge populations, policy makers, general physicians, and scientific communities. The present review aims to provide some information regarding the molecular and cellular mechanisms underlying SARS-CoV-2 infection.

## 1. Introduction—Summary of the COVID-19 Pandemic

Severe acute respiratory syndrome coronavirus type 2 (SARS-CoV-2) is a single-stranded, enveloped RNA virus that is the causative agent of coronavirus disease 2019 (COVID-19) [1,2]. On 30 December 2019, a cluster of patients with pneumonia of unknown etiology was observed in Wuhan, China, and this was reported to the World Health Organization (WHO) [3]. The virus was initially referred to as “novel coronavirus 2019” (2019-nCoV) by the World Health Organization (WHO), and the virus was subsequently given the official name of SARS-CoV-2 on 11 February 2020. The WHO declared COVID-19 a pandemic on 11 March 2020 [4]. As of today (21 October 2022), there have been about 631 million confirmed cases of COVID-19, with about 6.57 million deaths [5]. However, circumstantial evidence and different observations indicate that both the number of patients and the number of deaths may be several-fold higher than reported [6,7,8]. Thus, the COVID-19 pandemic has not merely been a matter of medical emergency; rather, it has affected almost all aspects of human life. Medical emergencies resulted in almost all countries; the hospitals were flooded with COVID-19 patients, and non-COVID-19 patients were not allowed to enter hospitals in many countries, with other types of medical emergencies left unattended. The economic impact has been tremendous: (1) there was a complete or partial blockade of almost all sorts of activities in different countries; (2) educational institutions were mostly closed for months or years; (3) travel bans were expanded day by day; (4) industries were closed; and (5) agricultural activities were highly compromised. Finally, human relations were in jeopardy, and even religious activities were contained. In many countries, such as India, patients who died of COVID-19 were discarded as garbage into rivers and other places [9].

The healthcare delivery system of most of the countries were unprepared to handle this pandemic. Accordingly, several approaches were taken to handle this pandemic, and many of these endeavors were not evidence-based. Many efforts were diverted to contain the pandemic by any possible means. This has been reflected scientifically in the publication of more than 305,546 scientific articles, in PubMed alone, about COVID-19 [10]. These publications have broadly discussed almost all aspects of COVID-19: (1) virology of SARS-CoV-2, (2) epidemiology of COVID-19, (3) pathogenesis of COVID-19, (4) strategies for prevention of infection with the virus, and (5) management modalities. To provide insights about these facts, there have also been extensive discussions about the immune response to SARS-CoV-2.

The present communication will provide attention to some specific aspects of immune response to SARS-CoV-2 that bear long-lasting implications for the management of any future epidemic or pandemic. This review will discuss: (1) the concept and nature of pathogenic versus protective immunity of SARS-CoV-2-infected patients; and (2) the scope and limitations of vaccines against SARS-CoV-2, with insight into the preventive and therapeutic implications of vaccines. 

## 2. The Role of Host Immunity during SARS-CoV-2 Infection and Genesis of COVID-19

Epidemiological analysis has shown that SARS-CoV-2 infects people of all classes, although specific groups may be more susceptible to this virus. However, there remains heterogeneity of healthcare delivery systems among countries and regions. In addition, the strategies for diagnosing SARS-CoV-2 were divergent between countries and even between different parts of same countries. Furthermore, the divergent socioeconomic conditions of individual countries played a vital role regarding the magnitude of SARS-CoV-2 infection. In order to develop insights into SARS-CoV-2 infection, there should be some discussion about the nature of the virus. SARS-CoV-2 is a betacoronavirus (an enveloped, single-stranded RNA virus) that shares 79% sequence identity with SARS-CoV and has 96% homology with the RATG13 coronavirus strain in bats [2,11]. SARS-CoV-2 virions attach to human cells via their densely glycosylated spike protein and bind with high affinity to the angiotensin-converting enzyme 2 (ACE2) receptor in human cells [12,13,14,15,16]. Transmembrane serine protease 2 (TMPRSS2) is a cell surface protein primarily expressed by endothelial cells in the respiratory and digestive tracts. As a serine protease, it is involved in the cleaving peptide bonds of proteins that have serine as the nucleophilic amino acid within the active site. Recent studies have shown that SARS-CoV-2 requires ACE2 (the primary receptor), as well as TMPRSS2, for entry into epithelial cells [17,18,19]. 

According to the WHO, SARS-CoV-2 can spread from an infected person’s mouth or nose in small liquid particles when they cough, sneeze, speak, sing, or breathe. When the virus enters the oropharyngeal compartment, host immunity has the predominant role regarding the handling of the virus. After the entry of SARS-CoV-2 into the nasopharyngeal compartment, SARS-CoV-2’s access to the host cell is mediated by the host cell receptor ACE2. Binding between the virus and host occurs between the virus spike (S) protein and the host ACE2 receptor. As discussed, the role of TMPRSS2 is also important during viral entry into the host. After the localization of the virus into the target cells, viral RNA is released. Viral RNA is replicated, and structural proteins are synthesized, assembled, and packaged in the host cell, after which viral particles are released. The process is ongoing and may lead to the spread of the virus from cell to cell within the same tissue as well as to other tissues. The role of host immunity is important in this context. The immunity of the host may be divided into two broad groups: (1) innate immunity and (2) adaptive immunity. If the cells of the innate immune system are properly activated, the virus may be localized in target tissues [20,21,22]. On the other hand, if the virus is recognized and processed by antigen-presenting cells (APCs), these APCs carry the immunogenic epitope of the virus. Presenting these epitopes with self MHC class II molecules in immunological synapses would induce effective adaptive immunity by producing specific antibody and cytotoxic T lymphocytes [23,24,25,26,27]. If the processes of innate and adaptive immunity progress properly following SARS-CoV-2 infection, the virus may be contained and complications may be avoided. The establishment of SARS-CoV-2 infection and progressive COVID-19 in apparently healthy individuals indicates that pattern recognition receptors (PRRs) of the nasal cavity or oropharyngeal mucosa of the infected hosts have not been properly activated by the various molecules of the pathogen-associated molecular pattern (PAMP) of SARS-CoV-2 [28,29,30,31,32,33,34,35,36,37,38,39,40,41]. This is not exceptional for SARS-CoV-2 and has been observed in many other microbial infections. The cellular and molecular mechanisms underlying immune evasion by microbial agents have been characterized for many viruses. Regarding SARS-CoV-2, some observational evidence has been discussed, but more work would be required to validate the assumptions. We and others have analyzed these events in subjects with chronic hepatitis B virus (HBV) infection in HBV-transgenic mice and patients with chronic hepatitis B [42,43,44,45,46,47].

Most host cells express RIG-1-like receptors, whereas toll-like receptors (TLRs) are expressed by cells involved in the host innate immune response. When the innate immune system and the parenchymal cells are correctly functioning, the entry or localization of the virus may not be effective in prolonging the infection [48]. However, when the innate immune system is weak, such as in aged people or people with comorbidities, the establishment of SARS-CoV-2 infection is feasible. This was epidemiologically suspected and clinically apparent in the response to SARS-CoV-2 infection over the entirety of 2020, as the virus was a novel virus and innate immunity effectively failed in aged and comorbid patients [49,50,51]. However, many apparently healthy individuals were also infected by SARS-CoV-2, and the interplay between host immunity versus viral pathogenicity remains to be elucidated in future studies. Additionally, tissue tropism of the virus is an important variable in this context. The virus has been detected mainly in the oropharyngeal tract. The localization of the virus in the pulmonary tissues with possible mobilization to extrapulmonary organs has also been shown [52,53,54,55,56,57].

## 3. Diverse Clinical Presentations of COVID-19 and the Role of Host Immunity

The disease caused by SARS-CoV-2 has been named COVID-19. Similar to most viral infections, COVID-19 patients exhibit diverse types of clinical presentations due to various reasons [58,59,60,61,62,63,64,65,66,67,68,69]. There are no specific clinical features of COVID-19, although the disease is usually characterized by fever, cough, loss of taste, and sore throat at the onset. In some patients, the disease progresses to a severe form. The National Institute of Health (NIH) has recently provided guidelines regarding clinical symptoms of COVID-19 (last updated 26 September 2022) [70]. As per those guidelines, the clinical features of COVID-19 may be grouped into five categories. However, there may be overlap between the different categories. Asymptomatic or pre-symptomatic infection with apparent COVID-like symptoms, but positive for SARS-CoV-2 by nucleic acid amplification test, represents the most benign form of COVID-19. Patients with mild illness can exhibit any of the following signs: fever, cough, sore throat, malaise, headache, muscle pain, nausea, vomiting, diarrhea, and loss of taste. However, these patients do not have breathing problems or dyspnea. When SARS-CoV-2-positive patients show evidence of lower respiratory pathologies and have an oxygen saturation of ≥94%, they are regarded as having moderate COVID-19. Patients with oxygen saturation of <94%, a ratio of arterial partial pressure of oxygen to fraction of inspired oxygen (PaO_2_/FiO_2_) <300 mm Hg, a respiratory rate >30 breaths/min, or lung infiltrates >50% are categorized as having severe COVID-19. Critical patients have respiratory failure, septic shock, and evidence of multiple organ failure. Thus, clinically, SAR-CoV-2 patients may be classified into one or overlapping categories mentioned above. The clinical features may show some divergency if there are comorbidities or if the patients are of special groups. The immune response to SARS-CoV-2 and some other host factors play variable roles in determine the category of a COVID-19 infection. However, it is yet to be affirmed whether the nature of the virus or the nature of the hosts plays the critical role in defining pathological status.

## 4. The Role of Local and Systemic Host Immune Responses to SARS-CoV-2

SARS-CoV-2 is the third coronavirus outbreak affecting human beings since the beginning of 21st century, occurring after the outbreaks of SARS-CoV and Middle East respiratory syndrome (MERS) CoV. SARS-CoV-2 enters via the ACE2 receptor and TMPRSS2, both of which are expressed in respiratory epithelia. In vitro cell culture studies have shown that human airway epithelial cell models are efficiently infected by SARS-CoV-2, and nose and upper airways are, therefore, the preferred sites for virus transmission and replication [71]. Direct interactions between SARS-CoV-2 and ciliated epithelial cells have also been shown by scanning electron microscopy [72,73]. The cytopathic effects of SARS-CoV-2 have been documented in cell culture systems, and they are proposed to occur in patients with COVID-19 [74,75,76]. Thus, SARS-CoV-2 is capable of inducing tissue damage. However, the relative role of the virus and other factors during the course of development of COVID-19 remains to be clarified. In this respect, it appears that direct cytopathic effects of SARS-CoV-2 may not be the only dominant aspect determining the extent of various pathological lesions in COVID-19 patients. Although SARS-CoV-2 has mostly been detected in nose and mouth mucosa, as well as in pulmonary tissues, several studies have reported that COVID-19 patients suffer from pathological lesions in the hematologic, cardiovascular, renal, gastrointestinal and hepatobiliary, endocrinologic, neurologic, ophthalmologic, and dermatologic systems [77,78,79]. Finally, multiorgan failure is the final assault on morbid COVID-19 patients. It is assumed that the host immune response plays a cardinal role in COVID-19 pathogenesis. Moreover, evidence has been piling up regarding the role of the immune system in SARS-CoV-2-related tissue damage during COVID-19 [80,81,82]. In addition, post-COVID syndrome and long COVID are two highly disturbing consequences of COVID-19 [83,84,85,86]. Taken together, extrapulmonary manifestations of COVID-19 along with multiorgan failure (MOF), post-COVID syndrome, and long COVID indicate that, in addition to the direct cytopathic effects of SARS-CoV-2, SARS-CoV-2-induced immune modulation as well as several other multifactorial factors may play important roles in determining the pathogenic features of COVID-19. Proper understanding of the cellular and molecular mechanisms underlying these events is extremely relevant to the appropriate management of COVID-19 and its complications.

## 5. The Pathogenic Role of Immune Response to SARS-CoV-2—Cytokine Storms and Severe COVID, Pneumonia, and Multiorgan Failure

The initial assumption that SARS-CoV-2 infection was similar to a flu-like infection could not be substantiated after time over the course of the pandemic, as many COVID-19 patients developed severe forms of illnesses with features of acute respiratory distress syndrome (ARDS) [87,88,89,90]. It is assumed that when the initial interaction between PAMPs of SARS-CoV-2 and PPRs of the infected hosts is unable to block the progression of SARS-CoV-2 and its localization in the upper respiratory tract, then infection is established in the upper nasal area. As a consequence, viral replication progresses, with inflammatory mediators mobilizing in this area. Subsequently, the virus is able to move toward the lower respiratory tract. Now, the fundamental question is related to SARS-CoV-2-induced pathogenesis and the role of the immune system. Initially, it was assumed that virus-induced cytopathic effects are mainly responsible for the clinical features of COVID-19. However, with the advent of severe COVID-19 and analysis of patients with pneumonia and MOF, there are various pieces of scientific evidence indicating that the virus can escape from the upper to lower respiratory tract, which may be responsible for the lethal inflammatory response observed in some patients, known as cytokine release syndrome (CRS) [91,92,93,94].

## 6. Possible Mechanisms Underlying Failure of Containing SARS-CoV-2 in Nasal Mucosa and Development of Serious Illnesses

### 6.1. Mechanisms Underlying Failure of Containing SARS-CoV-2 in Nasal Mucosa

The virus, SARS-CoV-2, may be contained in the nasal mucosa if the innate immune system is endowed with the proper capacity to identify and destroy the virus. Recognition of viral single-stranded RNA (ssRNA) and double-stranded RNA (dsRNA) as PAMPs by PPRs of the host determine the extent of innate immunity during viral infection. A notable PPR is retinoic acid-induced gene 1 (RIG-I)-like receptors (RLRs) [95]. Additionally, toll-like receptors (TLRs, mainly TLRs 3, 7, and 8) also act to identify the virus. This usually leads to the production of antiviral mediators, interferon (IFNs) [96]. The activation resulted from an interaction between PAMPs and PPRs induce type I/III interferons. In addition, other proinflammatory cytokines, such as tumor necrosis factor alpha (TNF-α) and interleukin-1 (IL-1), IL-6, and IL-18, are released to contain the virus. At the same time, it is expected that the cells of innate immunity (monocytes, macrophages, neutrophils, and dendritic cells (DCs)) secrete a series of proinflammatory cytokines. In the next phase, these cells should take part in the induction of an adaptive immune response to SARS-CoV-2. When the functions of different mediators and cellular components are dysregulated following SARS-CoV-2 infection, the impaired activity of innate immune system allows for viral persistency [97]. Growing evidence is pointing to an alteration in (IFN)-1-induced immunity in COVID-19 patients. Indeed, COVID-19 influences the host immune response and weakens IFN-1 function in response to infection. SARS-CoV-2 can act at the state of induction by inducing antagonism [98,99,100,101,102]. Other mechanisms related to failure of containment of SARS-CoV-2 in the nasal mucosa have been reported [103,104], and more insights will be generated in the future.

### 6.2. Mechanisms Underlying Development of Severe COVID-19 and Role of Immunity

The cellular and molecular mechanisms underlying the unrestricted release of these inflammatory factors are poorly understood, and several hypotheses have been proposed that must be addressed by evidence in the future. As of now, it seems that pyroptosis, a highly inflammatory form of lytic programmed cell death (apoptosis), remains in action during SARS-CoV-2 infection [105,106,107,108,109]. In addition to apoptosis and necrosis, there is another form of cell death called proptosis. This is a type of programmed cell death and may be triggered by various stimuli. Pyroptosis is inherently capable of clearing various bacterial, viral, fungal, and protozoan infections. However, if not all microbial agents are cleared by pyroptosis, there may be an extensive inflammatory response in the efferent limbs. This can take place in immune cells and is also reported to occur in keratinocytes and some epithelial cells. As a consequence of pyroptosis, the release of proinflammatory cytokines is triggered in COVID-19 patients, which possibly affects macrophage and lymphocyte functions [110,111,112,113,114,115]. This is, perhaps, responsible for peripheral lymphopenia in severe COVID-19 patients. The role of inflammasome is also related to development of severe inflammation in COVID-19 patients [111,112,113,114,115,116]. In addition to this, attention should be focused on the initial failure or suboptimal induction of innate immunity in the host during initial exposure to SARS-CoV-2 in order to allow for the development of proper insights into the cellular and molecular mechanisms underlying severe COVID-19.

As a summation, it appears that the primary focal point may be directed to an alteration in (IFN)-1-induced immunity in COVID-19 patients. Indeed, COVID-19 influences the host immune response and weakens IFN-1 function in response to infection. SARS-CoV-2 can act at the state of induction by inducing antagonism. As an effective type I IFN response is not found after SARS-CoV-2 infection, stimulation of other cells of the innate immune system, such as macrophages, dendritic cells, and neutrophils, occurs as the first line of defense. This paradigm is partially supported by the observation of massive macrophage infiltration in the bronchial mucosa in the lung autopsies of patients who died due to COVID-19 [117]. In line with this, recent studies suggest that excessive production of some cytokines, such as interleukin (IL)-6, may play a role in the case of progressive COVID-19 [118]. Thus, there is massive production of proinflammatory cytokines, and they form the basis of the so-called cytokine storm and induce damage to the lungs and other tissues.

## 7. Protective Immunity during SARS-CoV-2 Infection

Until now, we have mainly concentrated on pathogenic immunity during SARS-CoV-2 infection by outlining the realities for infection via the weak innate immunity and by citing the role of cytokine storms in the pathogenesis of severe and critical COVID-19. However, the impact of the immune response to SARS-CoV-2 may be viewed from another angle, such as the fact that large numbers of people have avoided infection or amounted immune responses that resulted in unreported asymptomatic infections. As of today, about 600 million people have been infected with SARS-CoV-2. In fact, the number may be several times more than that reported. Some investigators have predicted that the real number of cases may be four times that of the reported patients. Thus, about 2 billion people may have been infected by SARS-CoV-2 if the assessments of some epidemiologists are considered valid [6,7,8]. Now, there remains a query about the missing billions. The total population of the world is about 7.9 billion. If about two billion people (three to four times the reported cases) are infected by SARS-CoV-2, the missing 5 billion may have encountered the virus, but their immune response may not have allowed virus entry. Alternatively, perhaps the virus has entered, but there were no dominant symptoms. It is also possible that they have not encountered the virus at all. Within these possibilities, the extent and properties of protective immunity deserve discussion.

Additionally, the protective role of immunity is manifested by the nature of illnesses. There remain several concerns about the real extent of asymptomatic SARS-CoV-2 infection. In 2020, Ing et al. suggested that about 81% COVID-19 patients were asymptomatic [119]. In 2021, Ma et al. showed that about 40% of confirmed SARS-CoV-2-infected patients were asymptomatic by conducting a meta-analysis [120]. Additionally, considerable numbers of SARS-CoV-2-infected patients developed mild pathogenic features. Although the exact nature of protective immunity has not been properly declassified, it appears that in most SARS-CoV-2-infected persons, there is either induction of proper innate immunity or suppression of disease progression by virtue of the activities of immune cascades. Investigators have found that SARS-CoV-2 infection generates near-complete protection against re-challenge. This was first documented in rhesus macaques. Subsequently, there is limited evidence of reinfection in humans with previously documented COVID-19. These facts indicate the protective nature of SARS-CoV-2 infection in animals and humans [121], and, similarly, there is limited evidence of reinfection in humans with previously documented COVID-19 [122]. As the humoral immune response wanes following SARS-CoV-2 infection, strong memory B cell responses could not be shown. In fact, these are short-lived. However, memory T cell responses seem to be long-lived. Thus, proper assessment of memory T cells would provide insights into the cellular mechanisms underlying asymptomatic infection as well as mild infection [123].

Along these lines, vaccine-induced protection is also a paramount feature of protection against SARS-CoV-2 infection. Several types of vaccines have been developed to counter SARS-CoV-2 infection. Several vaccines, including mRNA, adenoviral-vectored, protein subunit, and whole-cell inactivated virus vaccines, have now been extensively used in several countries. The fundamental principle of vaccine development is based on the induction of neutralizing antibodies (NAbs) against the SARS-CoV-2 spike protein. Although there remain some diverse findings concerning the production of Nab by these vaccines, production of activated T lymphocytes has been recorded in most cases. Thus, protection against SARS-CoV-2 is expected after vaccination [124].

The above descriptive effects of vaccination are further evidenced by the data gathered after vaccinating the general population around the world. As shown in Figure 1, some dominant variables exposed over the past 30 months of the pandemic demand a crucial analysis of protective immune responses in SARS-CoV-2 infection. This would ultimately pave the way for controlling the pandemic in a scientific manner and also aid in handling future epidemics or pandemics. The fundamental observations that have provided insights into the protective immune response to SARS-CoV-2 are summarized below: 

### 7.1. During the Pre-Vaccinated Era

The immune systems of aged persons are more susceptible to infection. A higher rate of infections was observed in elderly populations (especially care facilities), and comorbidities exacerbated immune inadequacies. As a result, there were higher mortality rates observed in the elderly. As mentioned previously, the healthcare delivery system, national strategy to contain COVID-19, capabilities of the medical facilities to fight COVID-19, and socioeconomic conditions play essential roles in the containment of COVID-19. However, host immunity is one of the main determinants. Epidemiological analysis regarding the nature of SARS-CoV-2-infected patients in the pre-vaccinated era revealed that over the initial 12–16 months of the pandemic, most of the patients with severe COVID-19 were either aged people or people with comorbidities [125,126,127,128,129,130,131]. In other words, this population was somehow immune-compromised. There are undoubtedly several different causes underlying these findings. However, the general picture is more or less acceptable.

### 7.2. Insights into COVID-19 during the Post-Vaccination Era

The epidemiological pattern and nature of COVID-19 in patients has shown considerable alteration after administering two and three doses of COVID-19 vaccines during the recent ten months [132,133,134,135,136,137,138,139,140]. There has been a shift in the age of the patients suffering from COVID-19. Most infected patients in many countries were young people, and the COVID-related symptoms became milder, which is reflected in lower numbers of fatalities. The pandemic caused by SARS-CoV-2 is ongoing. However, the cases and deaths have been contained since a considerable proportion of the world population was vaccinated. 

## 8. Vaccine Breakthrough Infection

The advent of vaccines against SARS-CoV-2 has shown a tremendous positive implication for containment of SARS-CoV-2 infection and spread of the COVID-19 pandemic. Vaccines against SARS-CoV-2 induced considerable optimism following their introduction in late 2021. Short-term trials and clinical observations revealed that after complete vaccination (≥14 days after receipt of all recommended COVID-19 vaccine doses), protection against infection reportedly reached up to 95% [132,133,134,135,136,137,138,139]. However, along with time and following emergence of new variants of SARS-CoV-2, the limitations of the vaccine as a tool to contain the COVID-19 pandemic became evident. Several investigators have tried to unfold the possible mechanisms underlying the limited efficacy of the vaccines, especially in the era of Omicron variants [140,141,142,143,144]. Analyzing these factors may help us to progress and to develop more efficient vaccines against SARS-CoV-2. However, the preventive and therapeutic implications of a vaccine for the containment of SARS-CoV-2 and the control of severe infection will receive the attention of the scientific community as they work to develop better regimens of vaccines to combat the SARS-CoV-2 pandemic.

## 9. Handling a Pandemic with Properties of a Double-Edged Sword

SARS-CoV-2 is a novel coronavirus that has devasted the entire world in terms of all aspects of human livelihood. The economic impact of the pandemic has been tremendous and long-lasting. The social effects are difficult to digest and will persist for a prolonged period. The healthcare delivery systems of almost all countries were mostly broken, and several years will be needed for their restoration. Still, the pandemic is ongoing, with hundreds of thousands of cases confirmed and thousands of fatalities occurring every day. Several factors are responsible for the propagation of this pandemic, and numerous conditions can account for the millions of cases and deaths due to this pandemic. This article has touched upon only one of the factors that may be properly realized to fight the future pandemic, namely the dissection of immunity to SARS-CoV-2 infection. As soon as the pandemic broke out, several measures were taken to contain it. These included the use of masks, maintaining social distancing, lockdown of several cities and metropolises, closure of various business properties, and extreme restrictions on traveling, among others. These measures were mostly taken on an international basis for varying periods and time points. In addition, many countries employ their own regulations. 

Ultimately, vaccines against the virus were developed, representing a fundamental pillar of the exploration of immunity and the containment of microbial infection. The initial part of our discussion detailed the pathogenic role of the immune system during SARS-CoV-2 infection, which is not within the capacity of the healthcare delivery system, except in asking that the vulnerable not be exposed. However, this is difficult to implement, as it is not a social-friendly approach. The protective aspect of the immune response against SARS-CoV-2 may be addressed through vaccines. Not all microorganisms are committed to exposing herd immunity, as this is mainly based on the nature of the microbial organism [145]. It is a reality that the best vaccine against SARS-CoV-2 has yet to emerge. The commercially available vaccines against SARS-CoV-2 are able to contain the severity of the disease, but do not seem to protect against re-infection in many cases. The positive contribution of this epidemic to the scientific world is the rapid development of several forms of vaccines, including nucleic acid-based vaccines, and their recognition by regulatory authorities. At the onset of the pandemic, the world failed to propagate the importance of protecting vulnerable and immune-compromised individuals. As immunity is diverse in nature, the teachings of this pandemic should be borne in mind and implemented in all future emergencies [146,147]. Even with all the emerging queries regarding SARS-CoV-2 infection, the role of a humoral- or cell-mediated immune response to infection or vaccines remains one of most important fundamental pillars related to the containment of infections [148]. In addition to their protective effect, these immune responses are also related to reduced occurrences of mutations and severe infections [149]. Similarly, vaccines also impart protective capacities against infection by multiple mechanisms [150].

## 10. Queries from Scientists, Physicians, General Population, and Policy Makers Regarding SARS-CoV-2 and COVID-19

Almost three years have passed since the first reporting of SARS-CoV-2 infection. Still, there remain simple queries from different groups of the population. It is also unclear why vaccination would possibly reduce the severity of the disease. The lack of proper animal models of SARS-CoV-2 is a paramount limitation to resolve these issues. On the other hand, the general population expects to know how vaccines work. The traditional belief that vaccines are able to prevent infection with microbial agents is mostly accepted in the context of SARS-CoV-2. However, it is true that the infection is spreading at a slower pace. This can be properly addressed after we develop more insights into the mechanisms of the virus and its nature as well as the interaction between the virus and the host. However, evolving viral epitopes, the adaptive response of the vaccine, and the difference in mortality between a vaccinated population and an unvaccinated population may underlie these realities. One of the areas of study may be Japan. Most people from Japan have been vaccinated four times, and some people have received their fifth dose of the vaccine. They also use masks and most of the original social measures. Even then, Japan has recorded millions of new SARS-CoV-2 infections during the last year, after its population received two doses of the vaccine. Even after receiving four or five doses of vaccines and adhering to almost all public health measures, the daily new cases number about 100,000 in Japan. The daily mortality due to COVID-19 has been on the rise during the last two months. The fundamental factors related to these queries are linked to the interplay between SARS-CoV-2 and the host immunity of the infected subjects. More studies regarding the mechanisms of the cellular and molecular events underlying these factors are urgently needed.

## 11. Conclusions

COVID-19 represents an incredible pandemic, involving billions of people and causing the loss of trillions of US dollars. The economic implications of human loss and social disabilities cannot be numbered nor counted. The pandemic has been contained to some extent. Even then, there are hundreds of thousands of new SARS-CoV-2 infections daily, and nearly a thousand deaths. The possibility of the emergence of mutant strains and variants also remains. To handle these situations, proper insights into pathogenic and protective immunity against SARS-CoV-2 are required. Some of the cellular and molecular mechanisms underlying pathogenic and protective immunity have been realized, but the critical events have yet to be fully explored and supported by evidence. The COVID-19 pandemic has been contained by various means. There is also a need to ascertain how a future pandemic, “pandemic X”, may be contained. Additionally, social problems related to epidemics must be resolved. Why is there vaccine hesitancy? Why is there such heterogeneity in vaccine distribution? The pandemic is a global problem, and this generation has been confronted with its ugly face, but more scientific studies would help us to fight this as well as subsequent future emergencies. All of these considerations are dependent on proper insights into and sufficient details regarding the immune responses to SARS-CoV-2 in different populations.. However, the role of vaccination and the production of protective antibodies, along with antigen-specific T lymphocytes, will alter the paradigm of the pandemic. Most countries are now progressing rapidly to either a new normal or a normal life style. Although more insights into pathogenesis and the role of the vaccine would provide more insights for effective control of this pandemic, the baseline work has been accomplished during the last three years.

## Figures and Tables

**Figure 1 vaccines-11-00615-f001:**
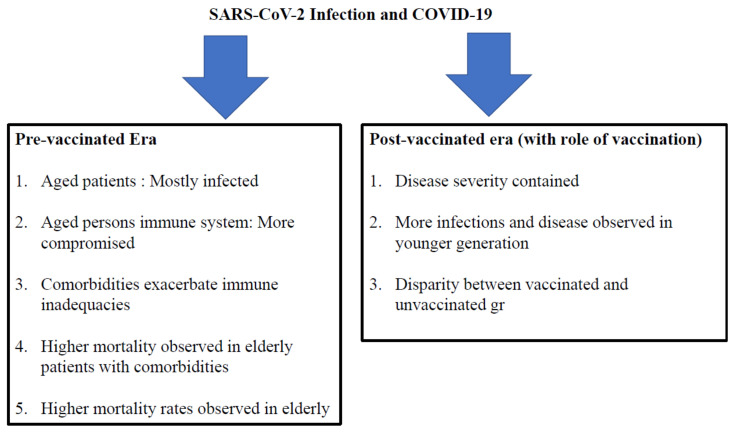
SARS-CoV-2 infections in different eras of the pandemic.

## Data Availability

Not applicable.

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
