# Peer review of "Cellular and Molecular Mechanisms of Pathogenic and Protective Immune Responses to SARS-CoV-2 and Implications of COVID-19 Vaccines"

_vaccines, 2023, doi:10.3390/vaccines11030615_

Round 1

Reviewer 1 Report (Previous Reviewer 3)

The authors addressed the concerns raised during the previous review. The revised manuscript is acceptable for publishing.

Author Response

Response to Reviewer 1

Thank you very much for your understanding.

Reviewer 2 Report (New Reviewer)

This is an interesting work, giving a lot of general information regarding SARS-COV-2 pathogenesis and immunity. Pathogenesis and especially evasion of immunity at the entry of the virus is well described. However the description of the immunity is general without enough description of the different parts of the immune system playing an important role against the virus. There is no mention of the antibodies or the cells, elicited by infection or the vaccines, that take part in the immunity against severe disease. There is a good description of the results in the populations but more details are needed concerning the results of vaccination or infection in the immune system that protect people from the virus. This part and especially molecular and cellular mechanisms are missing from this review entitled cellular and molecular mechanisms.

Author Response

Response to Reviewer 2

Query

This is an interesting work, giving a lot of general information regarding SARS-COV-2 pathogenesis and immunity. Pathogenesis and especially evasion of immunity at the entry of the virus is well described. However, the description of immunity is general without enough description of the different parts of the immune system playing an important role against the virus. There is no mention of the antibodies or the cells, elicited by infection or the vaccines, that take part in the immunity against severe disease. There is a good description of the results in the populations but more details are needed concerning the results of vaccination or infection in the immune system that protect people from the virus. This part and especially molecular and cellular mechanisms are missing from this review entitled cellular and molecular mechanisms.

Response

In order to address the concern of the honorable Reviewer, the following information has been incorporated into the article:

  1. The general immunity, as well as vaccine-induced immunity, and the role of antibodies as well as T lymphocytes, have been cited with reference. (Line 281-280, Reference 123-125)
  2. As mentioned by the honorable Reviewer, a comprehensive description has been provided about antibodies and T lymphocytes in the context of protection by vaccination (Line 282-290, Reference 126)
  3. Additionally, the role of vaccination has been emphasized in the final lines of the CONCLUSION.

All alterations have been made with Yellow markings.

Round 2

Reviewer 2 Report (New Reviewer)

The authors have added important information concerning immunity after infection or vaccinations, as they should according to the manuscript title.

On lines 276-279 they should correct as follows: as the...., strong memory B cell responses could not be shown.

Perhaps some more references in the literature concerning humoral or cell mediated immune response to infection or vaccines could be added to this review.

Author Response

Response to Comments of the Reviewer 2

Thank you very much for your constructive comments.

The responses are provided below.

All alterations shave been shown by yellow shading.

Comments of the Reviewer;

The authors have added important information concerning immunity after infection or vaccinations, as they should according to the manuscript title.

Response of the Authors:

Thank you very much for your understanding.

The query of the Reviewer:

On lines 276-279 they should correct as follows: as the...., strong memory B cell responses could not be shown.

Response of the Author

This has been corrected as advised.

The query of the Reviewer

Three more references have been added to substantiate the comment of the Review (Reference 148-150)

This manuscript is a resubmission of an earlier submission. The following is a list of the peer review reports and author responses from that submission.

Round 1

Reviewer 1 Report

The quality of the revised manuscript was greatly improved in part by the addition of new paragraphs

No more comments

Reviewer 2 Report

The manual script “Cellular and Molecular Mechanisms of Pathogenic and Protective Immune Responses to SARS-CoV-2 and Implications of 3 COVID-19 Vaccines”, as stated by the authors, is aimed to provide some information regarding the molecular and cellular mechanisms underlying 27 SARS-CoV-2 infection. However, the manuscript did not provide sufficient details of recent studies to fulfill this purpose. Tiny detailed cellular and molecular mechanism information is provided in the manuscript. The discussion about host immunity is simple and superficial.

The authors' major conclusion, “anti-SARS-CoV-2 vaccines and newly developed antiviral drugs would not be able to stand the test of time for eradication of the virus”, is no doubt overstated with insufficient information, and will be greatly disputable.

Reviewer 3 Report

The point of the manuscript appears to be to argue that the array of COVID-19 pathologies includes many that are due to aberrant immunity, and that a well-functioning host immune system should contain the virus (and disease) to the nasal and pulmonary tissues. However, the manuscript does not adequately describe innate and adaptive immune responses, nor immune responses that are protective. It lacks focus and does not provide the detail one would expect. This poorly written review does not add additional insight to the field and I do not recommend publication.

There are already hundreds of review articles describing various aspects of the immune responses to SARS-CoV-2 infection. The broad title of this review (Cellular and Molecular Mechanisms of Pathogenic and Protective Immune Responses to SARS-CoV-2 and Implications of COVID-19 Vaccines) implies that the reader will learn about protective responses, but only pathogenic responses are described.

The sole figure in the manuscript does not highlight any mechanisms or pathologies. It presents different era of the pandemic (and the role of vaccination) but the epidemiological evidence underpinning differences in the eras before and after vaccine availability is limited to two paragraphs. The discussion of vaccines is also limited. The authors touch on efficacy but conflate it with prevention of infection rather than prevention of severe disease. There is no information about engagement of the innate and adaptive immune responses after vaccination. Statements like “The scientific community is yet to get credible evidences regarding the role of vaccines for containing SARS-CoV-2 infection” completely overlook all the published clinical trials and additional publications describing the underlying mechanisms.

Finally, the authors lean on their publications in chronic Hepatitis B infections and animal models to validate their expertise in the field (mechanisms underlying host pathologies). Yet they assert that there is a “Lack of proper animals models of SARS-CoV-2”, neglecting numerous publications about mice, hamsters, ferrets, minks, nonhuman primates and even obese Ossabaw pigs (thinking of obesity causing aberrant immunity). Such biases are untenable in a fair scientific review of any subject.

Minor comments

Line 22, “property” should be plural.

Delete “on the other hand” as you have just said that the severity is related to host immunity.

Line 25, “pandemic, and divergent faces of COVI-19 have…”

Line 26, populations (plural)

Line 32, “severe acute respiratory syndrome coronavirus”

Line 51, reference for line about discarding dead patients in rivers.

Line 52, “were unprepared”

Line 54, recommend delete “in fact…” and use, “Many efforts were…”

Line 61, Remove the first sentence of this paragraph as it is not essential and actually detracts from the article. If the authors want to say an extensive review covering all aspects of the immune response is not practical and then they should they should just say that.

Line 73, “of individual countries ‘

Line 94, remove “and continual”

Line 97, I assume the authors mean if innate immunity is activated the virus could be localized (remove “not”.

Line 98 recommend “If viruses are localized..”

Line 111, why is it especially relevant for enveloped viruses? I recommend removing this as there is just as much reason to validate observational evidence for malaria as for COVID.

Line 112, the authors propose extending their experience with aberrant immune function and HepB to the interpretation of the observational evidence for COVID-19. If they just state that it would be easier to read rather than the last three sentences of the paragraph.

Line 113, ‘these studies have shown that’

Line 120, “firm localization” is imprecise, recommend “establishment of the virus is feasible”

Line 125 recommend deleting sentence “there remain paucities of information…”

Line 128, “Most relevant studies” is followed by references that focus on the lungs (including one for lung transplant). This doesn’t support the author’s assertion fairly. This is further exemplified by the following sentence that “convincing evidence of localization has not been demonstrated”. As the authors indicated in the first section their review is not intended to cover everything. Therefore they should remove this portion of the review and just stick with “Studies indicate a role of aberrant innate immunity in the development of COVID-19 and the localization of the virus in the pulmonary tissues with possible mobilization to extrapulmonary organs (then put references).

Line 170, I would recommend a title such as “The role of local and systemic host immune responses to SARS-CoV-2”. Direct cytopathic effects would be from disruption of cellular membranes as new virus emerges from infected cells, or the loss of essential protein production because the virus is using host ribosomes for viral protein production.

Line 237, there should be references supporting the statement that recent studies suggest IL-6 production plays a role in COVID-19 progression.

Line 260, recommend replacing “lacking strength of” with “weak”.

Line 262, recommend stating what the new angle is, e.g.: Large numbers of people have avoided infection or amounted immune responses that resulted in unreported asymptomatic infections.

Line 272, the assertion that 85-90% of infected persons develop asymptomatic infects should be supported with data or removed.

Line 276, the word “declassified” should be replaced with “elucidated”.

Line 285, Figure 1 is inadequate and subjective. The mortality statements are made without supportive data. The role of immunity isn’t indicated by the bullet points included in the figure.

Figure suggestion (although figures with cellular mechanisms of protection/pathogenesis are also needed)

Prevaccination Era: Aged persons immune system more susceptible to infection; More infections observed in elderly population (especially care facilities); Comorbidities exacerbate immune inadequacies; Higher mortality rates observed in elderly.

Postvaccination Era (with Role of Vaccination included): Disease severity contained; More infections and disease observed in younger populations; Disparity between vaccinated and unvaccinated.

Line 286, The use of the word kinetics in the title isn’t ideal.

Line 304, the section “Vital role of the immune response to COVID-19 and its reflection in the therapy of these patients” is woefully inadequate and should be deleted.  How can the role of immunity be substantiated by use of anti-inflammatory drugs? It’s analogous to saying the role of blood in wound healing is substantiated by the use of bandages.  There are documented changes as the immune system matures (from infants to children to adults to elderly) and the authors make no connection to this. It’s possible that the lack of documented infections in young children early in the pandemic was due to stronger engagement of the innate immune response rather than the adaptive immune response.

Line 309, the section “Vaccine breakthrough infections” is similarly inadequate and should be deleted. The generic comments from these two sections could be incorporated into other sections if they serve a purpose. The authors describe vaccine protection against infection but not protection against severe disease (which was the basis for authorization in many countries). There is no discussion on how the (different) vaccines affect the innate and adaptive immune responses. Nor is there a link between these mechanisms and those described for protective (or pathogenic) immune responses.

Line 312, vaccines were not the spearhead of public health responses; social distancing, masking, handwashing and isolation were the initial strategies until vaccines were developed and distributed. This statement is inaccurate and should be amended.

Line 322, there are many vaccines based on different production platforms available and in development. The authors should include a reference covering these and amend the sentence to reflect this.

Line 328, healthcare delivery has certainly been disrupted but it is a stretch to say that the delivery systems of almost all countries are broken. This statement is inappropriate.

Line 322, in this reviewer’s opinion the article has not describe the dissection of immunity in SARS-CoV-2 infection, either in reference to pathogenesis and protection, or in reference to innate and adaptive immune responses.

Line 343, the protective aspect of the immune response from most vaccines described to date relates to the production of IgG antibodies. It would have been nice to have that described in the review. While there are live attenuated vaccines in development that will likely confer some protection through other mechanisms and there may not be much information readily available the authors should have addressed this. Again, references to articles describing the mechanism of vaccines are lacking.

Line 359, see comment above about the credible evidence produced by the scientific community. This statement is clearly false. It may not be well translated into public messaging but there is ample evidence about the role of antibodies (from vaccines or as therapeutics) in limiting severe disease.

Line 363. The traditional belief about vaccines preventing infection needs to be put away and the notion that vaccines help the existing abilities of the immune system should be the narrative for policy makers. If there is a lesson from this pandemic that authors commenting on policy should convey it is better communication of science.

Line 367, the comments about the Japanese experience should have been supported with references. But again, the messaging is off point. The authors should have put this example in the context of evolving viral epitopes, the adaptive response of the vaccine, and the difference in mortality among a vaccinated population versus an unvaccinated population.

Line 376, the conclusion section states that most of the hypotheses provided in the article are based on epidemiological observation, but the authors do not clearly present any hypotheses. This is only the second time the word is used in the manuscript.

Line 385, there are already many cellular and molecular studies regarding COVID-19 disease and the statement that it is time to initiate such studies verges on being an insult to the many scientists who have changed the course of their studies and published those works in the last three years.